# In Vivo Acute Toxicity and Immunomodulation Assessment of a Novel Nutraceutical in Mice

**DOI:** 10.3390/pharmaceutics15041292

**Published:** 2023-04-20

**Authors:** Tatiana Onisei, Bianca-Maria Tihăuan, Georgiana Dolete, Mădălina Axinie (Bucos), Manuela Răscol, Gheorghița Isvoranu

**Affiliations:** 1The National Institute for Research and Development in Food Bioresources, Dinu Vintilă Street, No.6, 021102 Bucharest, Romania; 2Research Institute of the University of Bucharest—ICUB, 91–95 Spl. Independentei, 50567 Bucharest, Romania; 3Research & Development for Advanced Biotechnologies and Medical Devices, SC Sanimed International Impex SRL, 087040 Călugăreni, Romania; 4Department of Science and Engineering of Oxide Materials and Nanomaterials, Faculty of Applied Chemistry and Materials Science, University Politehnica of Bucharest, 011061 Bucharest, Romania; 5National Research Center for Food Safety, University Politehnica of Bucharest, 060042 Bucharest, Romania; 6National Institute of Pathology Victor Babeş—Bucharest, 99-101 Spl. Independenței, 050096 Bucharest, Romania

**Keywords:** immunomodulation, acute toxicity, nutraceuticals, animal studies, hydrolyzed collagen, *Vaccinium myrtillus*, *Echinacea purpurea*, royal jelly

## Abstract

Achieving and maintaining a well-balanced immune system has righteously become an insightful task for the general population and an even more fundamental goal for those affected by immune-related diseases. Since our immune functions are indispensable in defending the body against pathogens, diseases and other external attacks, while playing a vital role in maintaining health and modulating the immune response, we require an on-point grasp of their shortcoming as a foundation for the development of functional foods and novel nutraceuticals. Seeing that immunoceuticals are considered effective in improving immune functions and reducing the incidence of immunological disorders, the main focus of this study was to assess the immunomodulatory properties and possible acute toxicity of a novel nutraceutical with active substances of natural origin on C57BL/6 mice for 21 days. We evaluated the potential hazards (microbial contamination and heavy metals) of the novel nutraceutical and addressed the acute toxicity according to OECD guidelines of a 2000 mg/kg dose on mice for 21 days. The immunomodulatory effect was assessed at three concentrations (50 mg/kg, 100 mg/kg and 200 mg/kg) by determining body and organ indexes through a leukocyte analysis; flow cytometry immunophenotyping of lymphocytes populations and their subpopulations (T lymphocytes (LyCD3+), cytotoxic suppressor T lymphocytes (CD3+CD8+), helper T lymphocytes (CD3+CD4+), B lymphocytes (CD3−CD19+) and NK cells (CD3−NK1.1.+); and the expression of the CD69 activation marker. The results obtained for the novel nutraceutical referred to as ImunoBoost indicated no acute toxicity, an increased number of lymphocytes and the stimulation of lymphocyte activation and proliferation, demonstrating its immunomodulatory effect. The safe human consumption dose was established at 30 mg/day.

## 1. Introduction

Immunomodulation is a method of intrinsic or elicited regulation of the initiation, duration and level of the immune response (RI), and can be selective (when stimulation results in a response to one or more antigens) and non-selective (without directing the activity of stimulated cells to a specific antigen). The immune system (IS) offers a wide variety of amenable targets for immunomodulation in the treatment of infectious diseases, available for both prophylaxis and direct treatment [1,2]. Immunomodulatory agents cover a wide spectrum of natural and synthetic molecules that can be used for this purpose, including cytokines, hormones, glucocorticoids, peptides, microbial products, synthetic compounds, probiotics, nutrients, vitamins, minerals, plants and plant extracts, polysaccharides, vaccines and others [3,4,5]. The active role of nutrition and supplementation in supporting and maintaining immune homeostasis for different groups of individuals is currently highlighted in the literature, starting from topics with medical connotations such as immunonutrition to recently studied functional foods, dietary supplements and nutraceuticals [6,7,8,9,10,11]. The possible role of functional foods in modulating the (human) immune function is, however, at an early stage, and controversy over health claims will remain the rule rather than the exception until adequate biomarkers are identified to understand the individual responses and physiological and biochemical mechanisms that underlie the associations of nutrients in the diet [1,6,9].

Currently, the specialized literature contains constant and intensive concerns over the phytochemical and pharmacological study of plants and herbal remedies, for which the therapeutic properties in general, and among them the immunomodulatory ones, have been demonstrated [12,13,14,15,16,17]. The approaches are complex, from in vitro experimental studies to clinical studies that substantiate the use of some phytocomplexes and some minerals for prophylactic or curative purposes in food supplements with an immunomodulatory role. From this arises the concept of immunoceuticals, which refers to any nutraceuticals that are able to provide beneficial immunomodulatory actions that support and bolster the optimal immune system functioning [18].

Once this approach was expanded, insights about possible toxicological side effects emerged. This due diligence of scientific rigor was propelled by the differential regulatory approach administered to dietary supplements and nutraceuticals. Since the legislation was much more permissive with these types of products as compared to pharmaceuticals, numerous questions about their safety have been raised, especially since they were borderline approaching medical claims but were easily within reach of the general population, providing freedom in self-curing philosophies addressed by many. Therefore, the use of preclinical models for immunomodulatory studies became an indispensable part of the discovery and development process of any supplement or therapeutic agent. Although animal models only partially represent biomimetic models of immunostimulation or immunological suppression with the condition of heterogeneity, the usefulness of this approach is obvious, since the study of combinations of active principles is mandatory for demonstrating the safety and functionality of the developed food supplements; moreover, the multitude of combinations that can be developed means that the scientific literature only covers a narrow range of options.

Therefore, the main focus of this study was to assess the potentially acute toxicity and immunomodulatory properties of a novel nutraceutical with active substances of natural origin on animal models for 21 days.

## 2. Materials and Methods

### 2.1. Formulation Process and Rationale

The formula of the novel nutraceutical referred to as ImunoBoost (formula undergoing patent approval by the Romanian State Office for Inventions and Trademarks, registration number RO134617A0) is an innovative product that involves the use of optimized combinations between active principles of vegetable origin with antioxidant action and active substances of animal origin with immunomodulatory action, formulated for oral dosage by dispersion in watery solutions. The ImunoBoost product has the following composition: hydrolyzed bovine collagen 54%, egg yolk and lyophilized colostrum whey, lyophilized extract from aerial parts of *Echinacea (Echinacea purpurea)* with 4% echinacosides, blueberry fruit extract (*Vaccinium myrtillus*) with 40% proanthocyanidins and royal jelly. The product presented in a powder form was administered to the animal models in order to establish and recommend doses for safe human consumption, also talking into account the required correlation with literature findings and appropriate legislation.

### 2.2. Safety and Hazard Investigation of Novel Nutraceutical Formulation

#### 2.2.1. Bioburden Assessment

The bioburden of the formulated nutraceutical was assessed, determining the total number of viable aerobic microorganisms (TAMC), yeasts and molds (TYMC) (under the conditions provided by the European Pharmacopoeia volume 5, method 2.6.12 [19]). The working technique is based on the deep inoculation method, and 10 g of the product was added to 90 mL of buffer solution (peptone water with sodium chloride) at pH = 7 and homogenized. Serial dilutions (1/100, 1/1000) were prepared, using as the solvent the buffer solution. Next, 1 mL of the sample dilution was distributed into two sterile Petri dishes with a diameter of 90 mm. For TAMC, 15–20 mL of CaSoA culture media (BioMérieux, France) was melted, cooled to a temperature of 43 °C and added to each plate. The plates were incubated at 30–35 °C for 5 days. For the TYMC, 15–20 mL of SDA culture medium (BioMérieux, France) was melted, cooled to a temperature of 43 °C and added to each plate. The plates were incubated over a temperature range of 20–25 °C for 5 days. After incubation, colonies were counted only on the plates that showed microbial growth of a maximum of 250 for the total number of viable aerobic microorganisms and of a maximum of 50 colonies for the total number of yeasts and molds. The TAMC and TYMC are equal to the arithmetic mean of the number of CFUs (colony-forming units) recovered from the CaSoA and SDA culture media.

#### 2.2.2. Heavy Metals Analysis

Potentially hazardous toxic metals (As, Cd, Pb, Hg) were determined using an Agilent 8800 Triple Quadrupole ICP-MS supplied with an ASX500 autosampler (Agilent Technologies, Tokyo, Japan) in helium acquisition mode. The sample introduction system consisted of an integrated peristaltic pump set to 0.1 rps, which aspirated the sample into a concentric nebulizer. The operating conditions of ICP-MS were 1550 W of RF power, 1 L/min of carrier gas flow and 0.7 mL/min of He flow. The instrument was calibrated in the range of 0.5 to 25 µg/L for As, Cd and Pb, and from 0.1 to 10 µg/L for Hg. The calibration standards were prepared starting from 10 µg/mL standard solutions, purchased from Agilent Technologies. Prior to the sample measurements, an amount of 1 g of nutraceutical was weighed in a PFA digestion vessel, over which 8 mL of concentrated HNO_3_ (Suprapur^®^, Merck, Darmstadt, Germany) was added. The PFA tube containing the sample was subjected to a microwave-assisted digestion treatment (Ethos UP, Milestone Inc., Sorisole, Italy). The obtained solution was further quantitatively transferred to a 25 mL volumetric flask and subsequently diluted up to a mark with Milli-Q water (18 MΩ·cm).

### 2.3. Acute Toxicity and Immunomodulation Assessment

#### 2.3.1. Test Animals, Microclimate Conditions and Ethics Concerns

C57BL/6 adult mice (females) from the Animal Husbandry of “Victor Babes” National Institute of Pathology were used. The animals were kept under optimal conditions, with a temperature of 22 ± 2 °C, humidity of 55 ± 10%, artificial ventilation, lighting of 12/12 of light/dark, nourished and adapted ad libitum with granulated feed specific for mice and supplied with filtered and sterilized water. They were accommodated, differing by sex, in special cages with litter, respecting the density of individuals per cage. All mice were kept under a rigorous cleaning and hygiene program. Prior to blood collection, all animals were anesthetized with a ketamine/acepromazine cocktail, with ketamine at 100 mg/kg (ketamine 10%, Medistar Arzneimittelvertrieb Gmbh, Ascheberg, Germany) and acepromazine at 5 mg/kg (Calmivet Solution Injectable Acepromazine 5 mg, Vétoquinol SA, Lure, France). After the blood samples were collected, the animals were euthanized by dislocation of the cervical spine and organs of interest were harvested. The experimental procedures were performed in accordance with recognized principles of laboratory animal care in the framework of Directive 2010/63/EU on the protection of animals used for scientific purposes and Romanian national legislation. The study was approved by the Ethics Committee of “Victor Babes” National Institute of Pathology, Bucharest, Romania (No. 81/05.02.2020), and by the National Veterinary Sanitary and Food Safety Authority (No. 503/02.03.2020).

#### 2.3.2. Acute Toxicity

Three animals were used to test the acute toxicity of the novel dietary supplement. The dietary supplement was administered in a single dose (2000 mg/kg body weight) [20]. The single dose concentration was determined according to the scientific literature findings and in concordance with doses of similar products recommended for human consumption (30 g/day), multiplied by 100 and correlated with the mouse weight (23 g). This approach was considered viable because a lethal dose of this specific product cannot be obtained. The animals were monitored daily for physical and clinical symptoms and abnormal behavior, and weekly the weight was monitored. The animals were euthanized on day 14 and they underwent a pathological examination. The study protocol was established according to the Guidelines for Economic Cooperation and Development (OECD).

#### 2.3.3. Immunomodulation Assessment

For the evaluation of the immunomodulatory effect, 12 mice were used:(a)To the control group, only water was administered;(b)Group 1, where the dietary supplement was given at a concentration of 50 mg/kg of body weight;(c)Group 2, where the dietary supplement was given at a concentration of 100 mg/kg of body weight;(d)Group 3, where the dietary supplement was given at a concentration of 200 mg/kg of body weight.

The tested product was administered daily by gavage for 3 weeks. The clinical status and body weight of the animals were monitored throughout the experiments. At the end of the experiment, the animals were anesthetized as described in Section 2.3.1 and blood was collected from the retro-orbital plexus, then the animals were subsequently euthanized by dislocation of the cervical spine and organs of interest were harvested.

#### 2.3.4. Determination of Organs Index

After euthanizing the animals, the following internal organs were harvested and weighed: thymus, spleen, liver, kidneys. The weight index of the internal organs was calculated according to Formula (1):Organ index (mg/g) = organ weight (mg)/body weight (g) (1)

#### 2.3.5. Determination of Leukocytes

Blood samples were collected into EDTA tubes (Sarsted) to determine the number of leukocytes. The samples were analyzed using a HEMAVET 950 hematology analyzer (Drew Scientific). Before analyzing the samples, cycles of washing, background and control were carried out. Blood samples at room temperature were evaluated for blood clots and were analyzed according to species. The following parameters were determined: white blood cells (WBC), neutrophils (NE), lymphocytes (LY) and monocytes (MO).

#### 2.3.6. Isolation of Immune Cells

Immune cells were isolated from the spleens of mice. The spleen, a lymphoid organ, was harvested in a sterile environment and processed in order to obtain the suspension of immune cells; the spleen was mechanically processed, the resulting suspension was passed through a 70 μm sieve and centrifuged at 300× *g* for 5 min. After the centrifugation, the supernatant was discarded and 3 mL of lysis buffer (red blood cell lysis buffer) was added to the sediment and kept on ice for 5 min. After the lysis of the red blood cells, 7 mL of culture medium (RPMI 10% fetal serum) was added and centrifuged at 300× *g* for 5 min. After centrifugation, the supernatant was discarded and the sediment was recovered in the culture medium and passed through a 40 μm sieve to remove debris. The cell suspension was further centrifuged at 300× *g* for 5 min, then after the last wash, the cell sediment was resuspended in 1 mL of culture medium and the number and viability of the cells were determined by trypan blue staining.

#### 2.3.7. Cultivation of Immune Cells

Immune cells from the spleens of mice were cultured in the presence of mitogens at 37 °C in a 5% CO_2_ atmosphere for 24 h, 48 h and 72 h. The mitogens used were endotoxin from *E. coli*, lipopolysaccharides (LPS) (10 μg/mL) and concanavalin A (conA) (5 μg/mL).

Immune cells cultured for 24 h were labeled with fluorochrome-conjugated monoclonal antibodies in order to evaluate the lymphocyte activation, using an expression analysis of the early activation marker CD69. Splenocytes at 2 × 10^6^ cells/mL from the control group and from the treated groups were cultured in a 12-well plate in RPMI-1640 medium with 10% SFV and 1% antibiotic. After 24 h, the cell suspensions were washed and the monoclonal antibody was labeled according to the protocol below. Cells cultivated in the presence of mitogens for 48 h and 72 h were used for the lymphocyte proliferation assay.

#### 2.3.8. Determination of Cell Proliferation by 3-(4,5-Dimethylthiazol-2-yl)-5-(3-carboxymethoxyphenyl)-2-(4-sulfophenyl)-2H-tetrazolium Salt Reduction Assay

The CellTiter 96^®^ AQueous One Solution Cell Proliferation Assay (Promega) kit was used in determining the splenocyte proliferation index. The kit contained a tetrazolium compound [3-(4,5-dimethylthiazol-2-yl)-5-(3-carboxymethoxyphenyl)-2-(4-sulfophenyl)-2H-tetrazolium internal salt; MTS] and an electronically coupled compound (phenazine ethosulfate; PES). PES has chemical stability, and when combined with MTS forms a stable solution. The MTS compound is bio-reduced by the cells to the colored formazan, which is soluble in the culture medium. This conversion occurs in the presence of NADPH or NADH produced by dehydrogenases from metabolically active cells. From the control and treated groups, 200 µL of splenocytes 4 × 10^6^ cells/mL was cultured in a 96-well plate in RPMI-1640 medium with 10% SFV and 1% antibiotic. After 48 h or 72 h of cultivation, 100 µL of the cell suspension was pipetted and distributed to the other wells. Over this cell suspension, 20 µL of CellTiter 96^®^ AQueous One Solution reagent was pipetted into each sample. The reagent was added rapidly as it is extremely light-sensitive. The culture plate was incubated for 3 h at 37 °C with 5% CO_2_. Finally, the OD was determined using a plate reader at 492 nm against a reference wavelength of 620 nm. The proliferation index was calculated using Formula (2):Proliferation index = OD of the treated group/OD of the control group (2)

### 2.4. Immunophenotyping by Flow Cytometry

Lymphocyte immunophenotyping and a CD69 activation marker expression analysis were performed on cell suspensions obtained from the spleen by flow cytometry (BD FACSCanto II flow cytometer, BD FACSDiva v.6.1 program, BD Bioscience Inc., San Diego, CA, USA).

The flow cytometry analysis involves specifying the sets of monoclonal antibodies for the proposed investigations, marking samples with monoclonal antibodies, compensating for spectral overlaps, sample acquisition and a flow cytometer data analysis.

#### 2.4.1. Monoclonal Antibody Kits Used for Lymphocyte Immunophenotyping and Expression of Activation Marker CD69

By immunophenotyping, the lymphocytic parameters of cellular immunity were investigated based on specific surface markers. Thus, the following lymphocyte populations and subpopulations were identified and quantified: (a) T lymphocytes, characterized by the CD3ɛ+ phenotype, with the subpopulations: T-helper (CD3ɛ+CD4+) and T-cytotoxic/suppressive (CD3ɛ+CD8a+) (Table 1); (b) B lymphocytes, characterized by the CD19+CD3ɛ− phenotype; (c) NK cells, characterized by the NK1.1+CD3ɛ− phenotype (Table 1). All were acquired from BioLegend, San Diego, CA, USA.

The expression level of the CD69 marker was analyzed in lymphocyte cultures stimulated with lipopolysaccharide (LPS) and concanavalin A (ConA) (Table 2).

#### 2.4.2. Labeling of Samples with Monoclonal Antibodies Conjugated with Fluorochromes

For both types of determination, a surface marking protocol was used, carried out according to the Cell Surface Immunofluorescence Staining Protocol (BioLegend), as follows. For each sample, two tubes were used, one representing the negative control (unmarked sample), namely the sample marked with monoclonal antibodies, in which 100 µL of cell suspension (5–10 × 10^5^ cells/100 µL) was distributed by pipetting. For the Fc receptor blocking, 2 µL of TruStain FcX (anti-mouse CD16/32) (BioLegend) was added to each tube and incubated for 7 min on ice. Thus, by blocking Fc receptors, the non-specific fluorescent labeling was reduced. For the step of marking with fluorochromic monoclonal antibodies, in the tubes that constitute the marked sample, fluorochromic monoclonal antibodies were introduced by pipetting them in the concentrations indicated by the manufacturer, as follows:

-Lymphocyte immunophenotyping: 0.5 µL CD3ε Alexa Fluor 647; 0.5 µL CD8a Alexa Fluor 488; 1.25 µL CD4 PE/Cy7; 1.25 µL CD19 PerCP/Cy5.5; 1.25 µL NK-1.1 PE;-The CD69 marker analysis: 0.5 µL CD3ε Alexa Fluor 647; 0.5 µL CD8a Alexa Fluor 488; 1.25 µL CD19 PerCP/Cy5.5; 1.25 µL NK-1.1 PE; 5 µL CD69 PE/Cy7.

The tubes were homogenized and dark-incubated for 20 min on ice.

For the lysis of erythrocytes, 2 mL of 1× lysis solution (RBC Lysis Buffer, BioLegend) was distributed into each tube, homogenized and dark-incubated for 10 min at room temperature. Afterwards, the cells were washed in order to remove excess fluorochromes and centrifuged for 5 min at 350× *g*, the supernatant was removed and the resulting pellet was resuspended in 2 mL of washing buffer (Cell Staining Buffer, BioLegend). This step was performed twice, and finally the pellet was resuspended in 400 µL of wash buffer for the flow cytometry analysis.

#### 2.4.3. Compensation of Spectral Overlaps

The compensation of spectral overlaps aims to eliminate from the fluorescence channels the signals coming from other spectral channels and was achieved with the help of samples labeled with a single fluorochrome. The procedure involved marking the samples for clearance and their acquisition and analysis on the flow cytometer. Six compensation tubes were used as samples for compensation, one tube containing an unlabeled cell suspension, representing the unstained control (unstained control), and 5 tubes containing cell suspension in which a single monoclonal antibody used in the experiment was pipetted (stained control). The compensation samples were marked according to the protocol described in Section 2.4.2.

#### 2.4.4. Acquisition and Analysis of Compensation Samples by Flow Cytometry

For the lymphocyte immunophenotyping and CD69 activation marker expression analysis, the acquisition of compensation samples using the flow cytometer allowed the identification of distinct cell populations positive and negative for monoclonal antibody-bound fluorochromes, which was useful for the cytometry setup and automatic compensation calculation. By acquiring the events from the tube containing the unstained control (unstained control), the cell population of interest was virtually delimited (by creating a cytometric gate) (Figure 1), as were the negative events for the considered fluorochromes (Figure 2 and Figure 3).

In an FSC/SSC dot-plot histogram, a P1 gate containing the cells of interest was created.

Negative events for the fluorochromes used in the experiment can be visualized using the corresponding histograms (Figure 2 and Figure 3).

For the tubes containing the cell suspension labeled with individual fluorochromes, bimodal histograms were obtained, within which a P2 gate containing the positive events for the considered fluorochromes was created. For each individual tube, the voltages were adjusted in order to obtain the best possible separation between the positive and negative populations (Figure 4 and Figure 5).

After setting the voltages and adjusting the P2 gates, 5000 events were acquired for each tube, and finally the compensations were automatically calculated and applied both to the determinations made for lymphocytic immunophenotyping and for the analysis of the CD69 marker.

#### 2.4.5. Acquisition and Data Analysis for Lymphocyte Immunophenotyping and Expression of the Activation Marker CD69

The determinations were made at constant voltages, established in the compensation stage, and for each sample 100,000 events (lymphocyte immunophenotyping) and 500,000 events (expression of the CD69 activation marker) were acquired. Beforehand, the cytometer was calibrated (BD Cytometer Setup and Tracking Beads Kit, BD). The working document used for the data analysis is presented in Figure 6a–e.

The working document used for the data analysis is presented in Figure 7a–e.

In a “dot-plot”-type FSC-H/SSC-A histogram, the Singleti gate was constructed in which siglet events were included. This avoided the introduction of cellular aggregates into the analysis. From the gate containing the singlet events, the Ly gate (in an FSC-A/SSC-A cytogram) was constructed in which lymphocytes were isolated.

Histograms were constructed from singlet events to quantify cells positive for CD3ε, CD8a, CD19, NK1.1 and CD69 (Figure 7c–g, images from lymphocytes). CD3ε+ lymphocytes were obtained by intersecting the Ly population with CD3ε+ events delineated on the CD3ε count histogram (Figure 7c). Ts lymphocytes were obtained by crossing CD3ε+ lymphocytes with CD8a+ events delimited on the CD8a count histogram (Figure 7d). From the population of lymphocytes negative for CD3ε (virtually isolated with the help of the invert gate function applied in the CD3ε count histogram, Figure 7c) were isolated with the help of a quadrant-type dot-plot (CD8a/NK1.1), B lymphocytes (CD3ɛ−CD19+NK1.1− phenotype) and NK cells (CD3ɛ−CD19−NK1.1+ phenotype) (Figure 7h).

The expression level of the activation marker CD69 was assessed virtually on the total T lymphocytes (by intersecting the population of CD3ε+ lymphocytes with CD69+ events), Ts lymphocytes (by intersecting the population of Ts lymphocytes with CD69+ events), B lymphocytes (by intersecting the B lymphocyte population with CD69+ events) and NK cells (by crossing the NK cell population with CD69+ events).

### 2.5. Statistical Analysis

The obtained results were processed using the Microsoft Excel program. The results are presented as means ± standard deviations of the mean (SDs) (n = 3). The comparison between groups was made using the t-test with equal variance. We considered that the differences between the groups were statistically significant if *p* < 0.05 compared to the control group.

## 3. Results

### 3.1. Bioburden and Hazard Assessments

Evaluation of potential microbial hazards results obtained from the microbiological analysis of the ImunoBoost powder revealed absence of contamination. The total number of aerobic bacteria (TAMC) and the total number of yeasts and molds (TYMC) are expressed in CFU/g. The recorded results for both assays were <10 CFU/g (Table 3), implying that the sample presented no microbiological risk; therefore, the quality of the finished products was in accordance with values imposed by European Pharmacopoeia 5, standard ISO 4833-1/2014 and ISO 21527-2:2009.

The assay used for the determination of hazardous toxic metals showed good linearity for all elements in the selected ranges, with correlation coefficients greater than 0.999.

The concentrations of the sample are showed in Table 2 and are expressed as the mean values of the triplicate analysis, followed by the standard deviations. Toxic metals such as Cd, Pb and Hg were compared with the maximum levels according to Commission Regulation (EC) No. 1881/2006. Lead, cadmium and mercury are the only metals that are specifically regulated for dietary supplements, with maximum levels of 3 mg/kg for Pb, 1 mg/kg for Cd and 0.1 mg/kg for Hg. Arsenic, on the other hand, is not specifically regulated for dietary supplements, but we took into consideration the smallest value that is specified in EC No. 1881/2066, namely 0.1 mg/kg [21].

As we can see from Table 4, our sample did not show alarming values for heavy metals. Although arsenic, cadmium and lead were present in the samples in concentrations of 0.0312 ± 0.12 mg/kg, 0.0171 ± 0.03 mg/kg and 0.0201 ± 0.08 mg/kg, respectively, these values do not exceed the maximum regulated limits according to Commission Regulation (EC) No. 1881/2006, while mercury was not detected in any form, being below the instrumental detection limit.

### 3.2. Acute Toxicity

Fourteen days after the administration of a single dose of ImunoBoost (2000 mg/Kg body weight), no physical or clinical signs were observed regarding the health of the animals. The animals were evaluated daily. The body index of the control group was 23.27 ± 0.4 after 14 days, while for the acute dose group it was 22.20 ± 1.1, indicating a 4.59% decrease. Moreover, after the animals were euthanized, no pathological lesions were observed in the internal organs. In addition, all animals survived and the slight variations in weight observed were not considered significant in comparison to the control group (Table 5, Figure 8).

After the internal organs were harvested and weighed, the weight index/organ ratio was calculated according to Formula (1) from Section 2.3.2. No significant differences were observed in the growth of internal organs in animals that were tested for an acute dose (Table 6).

Fourteen days after the administration of the tested product dose, a significant increase in the number of leukocytes was observed compared to the control group (Table 7, Figure 9), from 7.8 ± 1.52 to 12.67 ± 0.49. The increased total leukocyte count (WBC) may indicate a stimulatory effect on hematopoietic stem cells.

### 3.3. Evaluation of the Immunomodulatory Effect

The experiment was conducted over a period of 21 days, during which time the animals received water and food ad libitum. Three doses of 50 mg/Kg body weight, 100 mg/Kg and 200 mg/Kg ImunoBoost were administered to three mouse groups. The body weight (g) was measured weekly. Group 1 registered 21.73 ± 0.4 g after 21 days, group 2 22.17 ± 0.6 g and group 3 22.33 ± 0.4 g. The control group weighed 23.27 ± 0.4. A decrease of less than 5% was observed for all groups; therefore, these slight variations were not considered significant in comparison to the control group (Table 8 and Figure 10).

An assessment of internal organ weight index led to the conclusion that no significant differences were observed in the growth of internal organs in animals that received the tested product compared to the control group (Table 9).

After 21 days of receiving the tested product, an increase in the number of leukocytes compared to the control group (Table 10, Figure 11) was observed. The highest WBC (K/μL) value (10.59 ± 3.2) was observed for group 1, which received 50 mg/Kg of ImunoBoost. The WBC value for group 2 was 9.41 ± 1.3, while it was 9.83 ± 1.4 for group 3. The increased total leukocyte count (WBC) may indicate a stimulatory effect on hematopoietic stem cells. Slightly increased lymphocytes values were also observed (group 1 LY 7.80 ± 2.1, group 2 7.05 ± 0.9, group 3 7.08 ± 0.7, as compared to LY 5.89 ± 1.4 for the control group).

### 3.4. Lymphocyte Immunophenotyping and Analysis of Activation Marker CD69 Expression

The evaluation of the lymphocyte populations and subpopulations in the spleens of mice receiving ImunoBoost did not reveal changes in the percentages of T lymphocytes (LyCD3+), cytotoxic suppressor T lymphocytes (CD3+CD8+), helper T lymphocytes (CD3+CD4+), B lymphocytes (CD3−CD19+) and NK cells (CD3-NK1.1.+). In addition, there were no differences from the control group in terms of the Th/Ts ratio (Table 11).

The stimulation of the immune cells in culture for 24 h with LPS resulted in increased activation. The total T lymphocyte (LyCD3+CD69+) counts were 11.6 ± 4.1% for the control group, 12.2 ± 2.5% for group 1, 28.9 ± 5.2% group 2 and 29 ± 4.5% group 3. The highest stimulation was observed in group 3, which received 200 mg/Kg. The Ts lymphocyte (CD8+CD69+) counts were 9.2 ± 2.1% for the control, 8 ± 1.9% for group 1, 27 ± 3.9% group 2 and 26.6 ± 3.3% for group 3. In this case, both group 2 and 3 showed significant stimulation, with group 2 achieving the highest values. The B lymphocyte (CD3−CD19+CD69+) count for the control was 40.3 ± 2.5%, for group 1 was 35.9 ± 6.3, for group 2 was 48.8 ± 4.3 and for group 3 was 44.9 ± 4.9. Groups 2 and 3 showed significant stimulation, with group 2 achieving the highest values. The NK cell (CD3-NK1.1.CD69+) value for the control was 18 ± 1.2, for group 1 was 16.7 ± 4.1, for group 2 was 33.1 ± 3.9 and for group 3 was 31.3 ± 4. Group 2 achieved the highest stimulation values. Therefore, the immune cells from mice in groups 2 and 3, which received higher doses of ImunoBoost, showed a significant increase in the expression of CD69, an early activation marker expressed on lymphocytes (Table 12, Figure 12).

The stimulation of the immune cells in culture for 24 h with conA resulted in increased activation as well. The total T lymphocyte (LyCD3+CD69+) counts were 48.7 ± 8.4 for the control group, 50.5 ± 3.2 for group 1, 67.6 ± 10.1 group 2 and 72.1 ± 2.3 group 3. The highest stimulation level was observed in group 3, which received 200 mg/Kg. The Ts lymphocyte (CD8+CD69+) counts were 58.7 ± 9.9 for the control, 53.2 ± 3.5 for group 1, 70 ± 0.7 for group 2 and 74 ± 1.1 for group 3. In this case, group 3 showed significant stimulation and achieved the highest values. The B lymphocyte (CD3−CD19+CD69+) count for the control was 45.1 ± 0.5, for group 1 was 47.3 ± 5.7, for group 2 was 66.6 ± 2.5 and for group 3 was 76.7 ± 0.7. Group 3 presented significantly higher values. The NK cell (CD3-NK1.1.CD69+) count for the control was 27 ± 1.4, for group 1 was 22.8 ± 2.9, for group 2 was 38.2 ± 9.8 and for group 3 was 29.36 ± 4. Group 2 achieved the highest stimulation values. Similar to LPS stimulation, the immune cells from mice in groups 2 and 3, which received higher doses of ImunoBoost, showed a significant increase in the expression of CD69, an early activation marker expressed on lymphocytes (Table 13, Figure 13).

As a general observation, group 2 obtained greater stimulation rates with LPS, while group 3 performed better under stimulation with conA.

In the case of splenocyte cultivation in the presence of LPS, a significant increase in proliferation capacity was observed only in the case of the high dose of the product (group 3—0.155 ± 0.023) compared to the control group (0.113 ± 0.004) (Table 14, Figure 14) after 72 h. In the other two groups, the proliferation capacity of the splenocytes was similar to that of the control group.

The stimulation of lymphocyte proliferation via modulation with conA showed that the splenocytes isolated from animals receiving the novel nutraceutical at the dose of 200 mg/kg proliferated almost two times more (0.269 ± 0.049) than those from the control group (0139 ± 0.01) (Table 15, Figure 15).

## 4. Discussion

The expansion of the global nutraceuticals market, which includes immunoceuticals, surpassed 400 billion USD in revenue in 2021, with its growth being estimated to increase even higher in the immediate future [18]. This indicates how stringent the need for maintaining a balanced immune system has become. Since the innate and adaptive responses provide immunity, harboring and boosting their function via active or passive protection is essential. The benefits of immunity fine-tuning by nutrition include inflammation management, gut health, disease counteraction and prolonging general health [1,2,6].

The health benefits associated with the consumption of hydrolyzed collagen [22,23,24,25,26,27] as well as other animal-origin active substances such as whey [28,29,30,31] and royal jelly [32,33,34,35] promote them as ideal ingredients for immunomodulation purposes. Moreover, the antioxidant properties of *Vaccinium myrtillus* [36,37,38,39] and *Echinacea purpurea* [40,41,42,43], coupled with their anti-inflammatory and antimicrobial effects, sustain their use for promoting immunomodulation and homeostasis.

The novel nutraceutical ImunoBoost (formula undergoing patent approval by the Romanian State Office for Inventions and Trademarks, registration number RO134617A0) is an innovative product that involves the use of optimized combinations between active substances of vegetable origin with antioxidant action such as *Vaccinium myrtillus* and *Echinacea purpurea,* as well as active substances of animal origin with immunomodulatory action. The formulation was optimized for oral dosage via dispersion in watery solutions for several reasons, such as bioavailability, fast absorption and consumer compliance in terms of easily embedding into the consumer’s lifestyle.

The potential hazards of microbiological and chemical origins were evaluated. The results obtained for the total number of aerobic bacteria and total number of yeasts and fungi were in compliance with the values indicated as safe for consumption by the ISO 4833-1/2014 and ISO 21527-2:2009 standards.

As for the heavy metal content, our sample did not show alarming values. The values obtained for arsenic, cadmium and lead did not exceed the maximum regulated limits according to Commission Regulation (EC) No. 1881/2006, while mercury was not detected, being below the instrumental detection limit.

The assessment of potential hazards is particularly important to this nutraceutical, since both vegetal and animal-origin ingredients were used, both being prone to harboring intrinsic hazards. Moreover, numerous reports over the years have raised concerns regarding correlations between the beneficial effects of nutraceuticals and their safety, compliance and transparency with regulatory demands [44], as well as in minimizing adulteration during production and commercialization [45,46]; therefore, safe consumption is considered the gateway to efficacy.

The acute toxicity was evaluated for 14 days after the administration of a single dose of the ImunoBoost nutraceutical to C57BL/6 adult mice (females). After administering a 2000 mg/Kg single dose, the body weight of the mice was evaluated at 0 days, 7 days and 14 days. The results obtained indicated that no significant weight loss, physical or clinical signs or pathological lesions were observed in comparison with the control group. All tested animals survived.

The organ weight can be a sensitive end-point indicator of an effect of an experimental compound, as significant differences in organ weight between treated and untreated (control) animals may occur in the absence of any morphological changes [47]. The assessment of the thymus, spleen, liver and kidney indexes indicated a slight increase in weight compared to the control group but no significant differences were noted. The evaluation of total numbers of leukocytes (WBC), lymphocytes (LY), neutrophils (NE) and monocytes (MO) after 14 days presented a significant increase in the number of leukocytes observed compared to the control group. The increased total leukocyte count (WBC) may indicate a stimulatory effect on hematopoietic stem cells.

Acute toxicity evaluation requires correlations between several markers, such as physiological parameters (weight, water intake), morphological aspects (abnormalities at organ level) and hematological markers (leukocytes, lymphocytes, neutrophils and monocytes) corroborated with behavioral changes observed during treatment in order to disseminate potential toxicological routes that stem from these base line assessments [48]. The results obtained after the administration of ImunoBoost indicated no acute toxicity signs and compliance with OECD guidelines regarding acute doses. Since we evaluated the LD50 at 2000 mg/kg and all mice survived, according to the OECD we can conclude that no lethal dose can be determined and no circumstances indicate the need to increase the dose to 5000 mg/kg, as presented by the guidelines for exceptional cases [48]. The literature reports present similar findings, with no acute toxicity determined for either of the ingredients used for obtaining the ImunoBoost formula [49,50,51,52,53,54]. Even if the results are promising, the reduced number of mice (n = 3) used for the toxicity assessment is one of the limitations of this study.

The immunomodulatory effect was evaluated over 21 days of administrating three doses of 50 mg/Kg body weight, 100 mg/Kg and 200 mg/Kg ImunoBoost to three mouse groups. The body weight (g) was measured weekly for all groups and less than 5% variations were observed by comparison with the control group. Similar results with no significant variations were obtained for the organ indexes as well, although a slightly increased number of leukocytes and lymphocytes compared to the control group was determined. This may have been due to a stimulatory effect on hematopoietic stem cells, the caging environment and stress exposure, all factors that may prime mice to respond differently to immune challenges [55].

The immunophenotyping of lymphocyte populations and subpopulations of mice after receiving ImunoBoost pointed out no significant changes in the percentages of T lymphocytes (LyCD3+), cytotoxic suppressor T lymphocytes (CD3+CD8+), helper T lymphocytes (CD3+CD4+), B lymphocytes (CD3−CD19+) and NK cells (CD3−NK1.1.+). The stimulation of immune cells for 24 h with both LPS and conA resulted in increased expression of CD69, especially for mice in group 2 (100 mg/Kg) and group 3 (200 mg/Kg).

The proliferation capacity of B and T lymphocytes was evaluated via modulation with mitogens such as LPS and conA. As we know, concanavalin A triggers T lymphocytes by directly interacting with their receptors for activation [56]; therefore, on splenocytes isolated from mice group 3 receiving the novel nutraceutical at the dose of 200 mg/kg, the proliferation values were almost two times higher than those from the control group.

Several literature findings indicate a positive immunomodulatory effect if the tested product can present modulatory properties of either innate or adaptative immunity via direct or indirect mechanisms [57,58,59,60]. More specifically, activation of macrophages, NK cells, T-cells and B-cells. Our findings indicate that without stimulation, no immunological modulation markers can be disseminated. On the other hand, once stimulation with LPS and ConA was applied, increased CD69 expression was observed, especially for group 3 (200 mg/Kg). Since CD69 is considered a metabolic gatekeeper [61], these results present promising outcomes regarding the capacity of ImunoBoost to stimulate the immune system, having traits of a non-specific immunomodulating agent.

As we evaluated the effect of ImunoBoost on mice from 14 to 21 days, we demonstrated that our product presents no acute toxicity when administered by gavage; yields good immunomodulatory effects, especially for doses of 50 mg/Kg; and provides higher rates of immunostimulation from 100 mg/Kg to 200 mg/Kg, providing an overall health-beneficial impact. Keeping this in mind, we estimated a safe human consumption dose range of 15–30 g of ImunoBoost/day for the general population (healthy subjects). Alongside a well-balanced diet, this type of nutraceutical can act as an adjuvant in strengthening the organism’s capacity to fight against immunity disruptors such as seasonal flue pathogens and can help boost immunity via its antioxidant potential, which can propel the normal energy-yielding process of the metabolism.

## 5. Conclusions

Immunomodulation, immune homeostasis and immunostimulation via nutrition have become focal points in the development of nutraceuticals and immunoceuticals in the last 10 years. Such products modulate and arm the immune system by keeping them in a highly prepared state against any threat. This study showed that the novel nutraceutical referred to as ImunoBoost, with natural bioactive substances in its formula, presents no acute toxicity, can increase the number of lymphocytes and promotes lymphocyte activation and proliferation, demonstrating its immunomodulatory effect. Although the results show promising effects, the study’s limitations such as the low number of animals tested and the relatively short period of exposure to the product should be considered. Further testing is necessary in order to validate the obtained results.

## Figures and Tables

**Figure 1 pharmaceutics-15-01292-f001:**
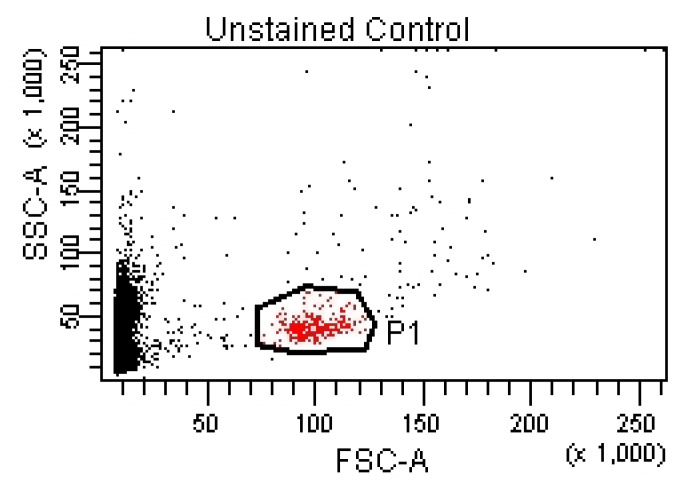
FSC/SSC dot-plot with isolated cells of interest (P1).

**Figure 2 pharmaceutics-15-01292-f002:**
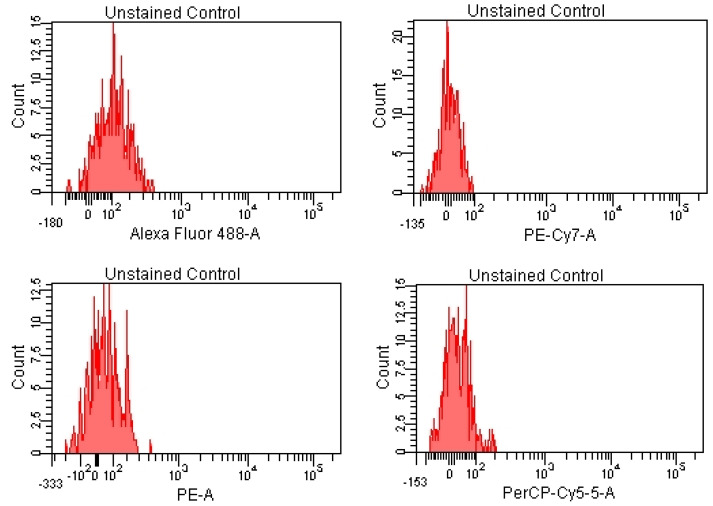
Unlabeled cell suspension (no specific fluorescence signals)—blue laser stimulation.

**Figure 3 pharmaceutics-15-01292-f003:**
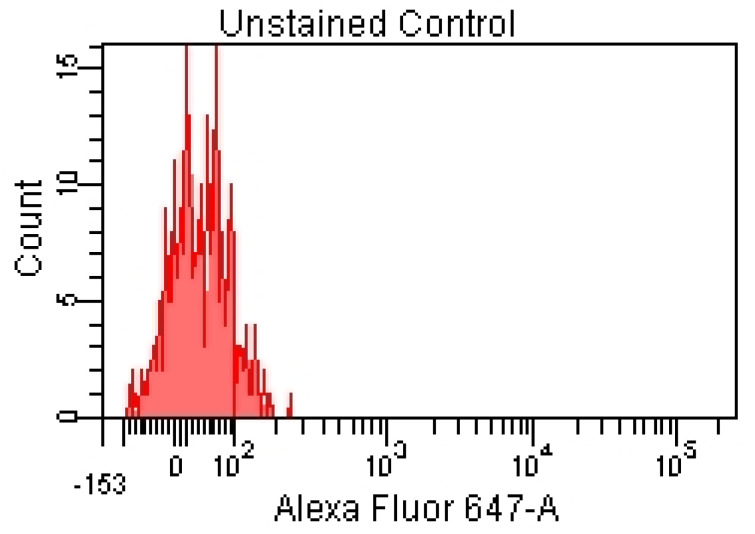
Unlabeled cell suspension (no specific fluorescence signals)—red laser stimulation.

**Figure 4 pharmaceutics-15-01292-f004:**
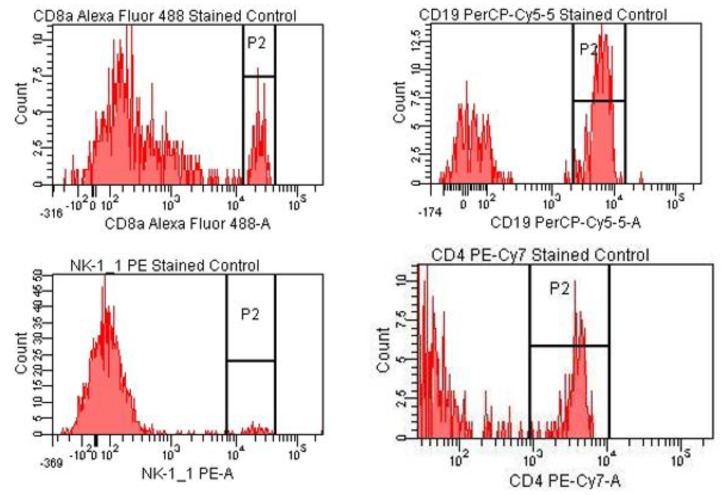
Labeled cell suspension—blue laser stimulation.

**Figure 5 pharmaceutics-15-01292-f005:**
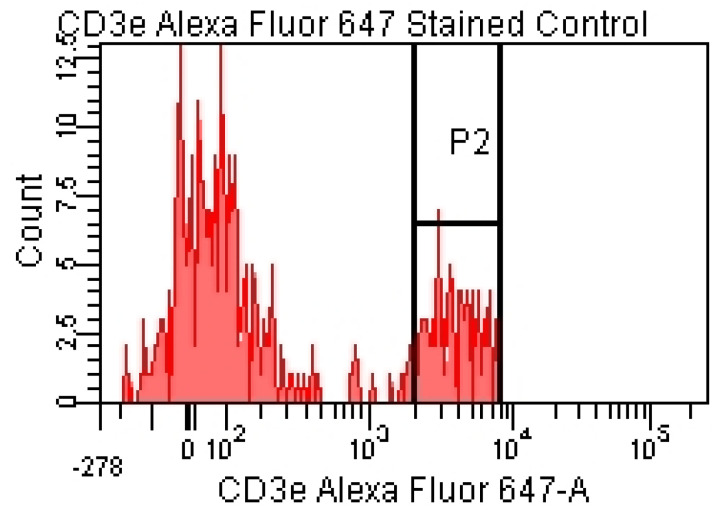
Labeled cell suspension—red laser stimulation.

**Figure 6 pharmaceutics-15-01292-f006:**
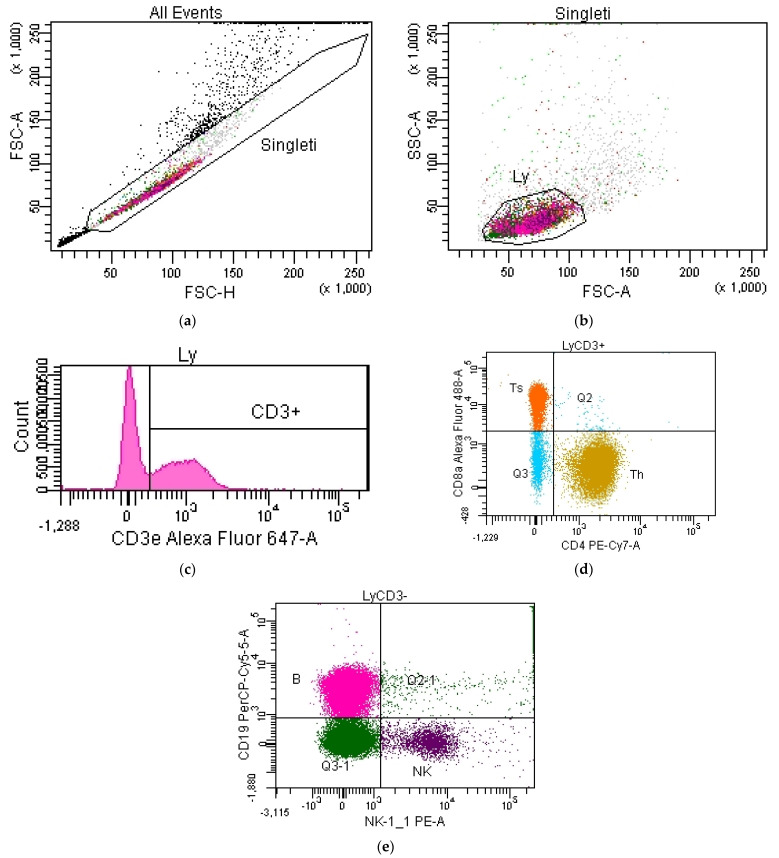
(**a**) Selection of singlet events. In an FSC-H/SSC-A dot-plot, the Singleti gate was constructed in which siglet events were included. The introduction of cellular aggregates into the analysis was avoided. (**b**) Selection of the lymphocyte population. From the gate containing the singlet events, the Ly gate (FSC-A/SSC-A) was constructed in which lymphocytes were isolated. (**c**) Selection of CD3ε+ lymphocytes. Using a histogram (CD3ε/Count), the population of CD3ε+ lymphocytes (total T lymphocytes) was isolated. The CD3ε- lymphocytes were virtually isolated using the ”invert gate” function. (**d**) Selection of Th and Ts lymphocytes. From the T lymphocytes (CD3ε+), using a CD4/CD8a dot-plot with a quadrant, we isolated the Th (helper) lymphocyte subpopulations with the phenotype CD3ɛ+CD4+CD8a− and Ts subpopulations (suppressor/cytotoxic) with the phenotype CD3ɛ+CD8a+CD4−. (**e**) Selection of B lymphocytes and NK cells. From CD3ε-negative lymphocytes, B lymphocytes (phenotype CD3ɛ−CD19+NK1.1−) and NK cells (phenotype CD3ɛ−CD19−NK1.1+) were isolated.

**Figure 7 pharmaceutics-15-01292-f007:**
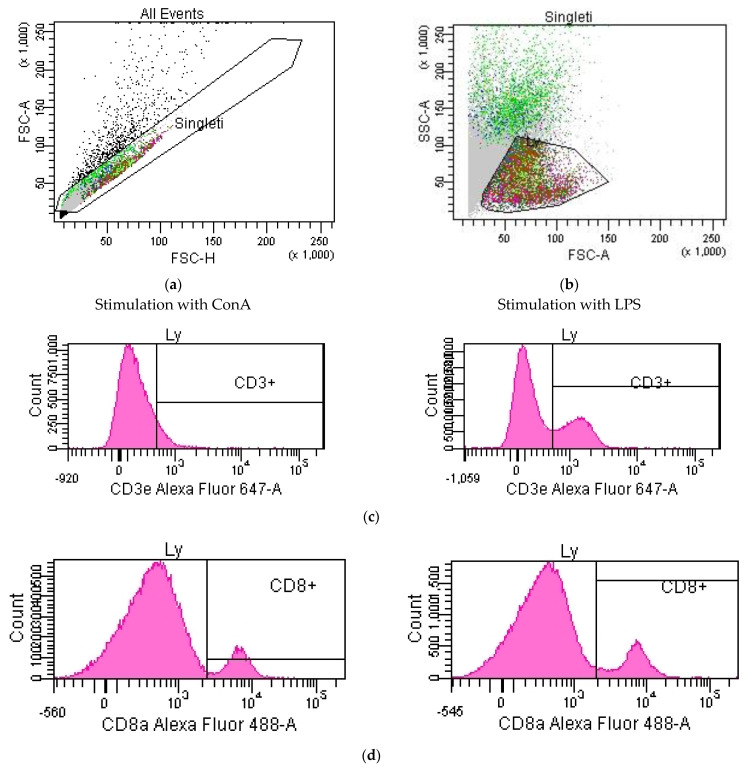
(**a**) Selection of singlet events. (**b**) Selection of the lymphocyte population. (**c**) Selection of lymphocytes of CD3ε+. (**d**) Selection of lymphocytes of CD8a+. (**e**) Selection of B lymphocytes. (**f**) Selection of NK cells. (**g**) Selection of lymphocytes of CD69+. (**h**) Selection of B lymphocytes and NK cells.

**Figure 8 pharmaceutics-15-01292-f008:**
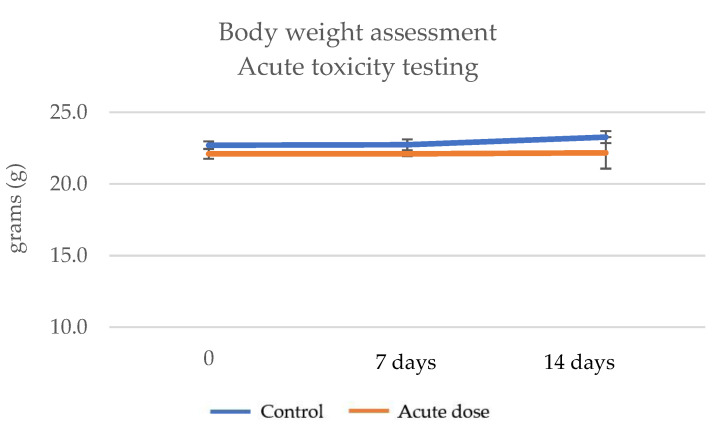
Body weight assessment—acute toxicity testing.

**Figure 9 pharmaceutics-15-01292-f009:**
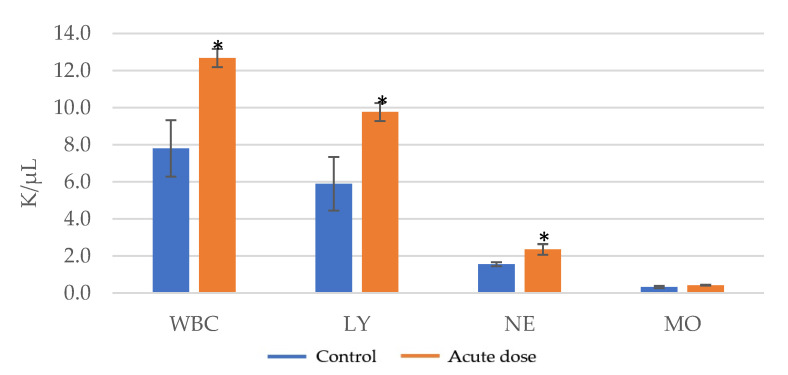
Assessment of leukocyte numbers in acute toxicity testing for the total numbers of leukocytes (WBC), lymphocytes (LY), neutrophils (NE) and monocytes (MO) in mice; * = *p* value < 0.05.

**Figure 10 pharmaceutics-15-01292-f010:**
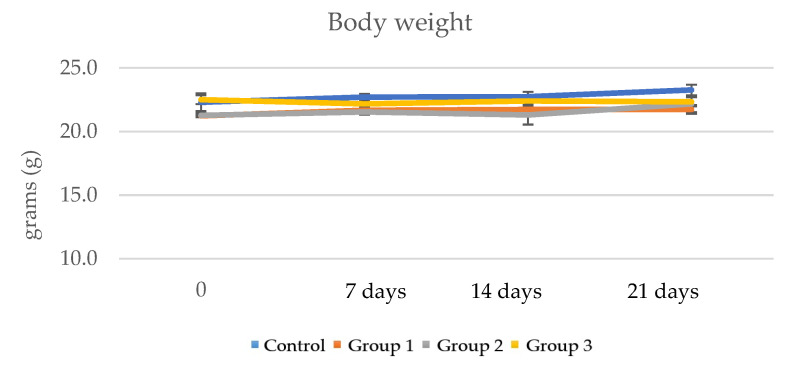
Body weight assessment of animals that received the product for 21 days.

**Figure 11 pharmaceutics-15-01292-f011:**
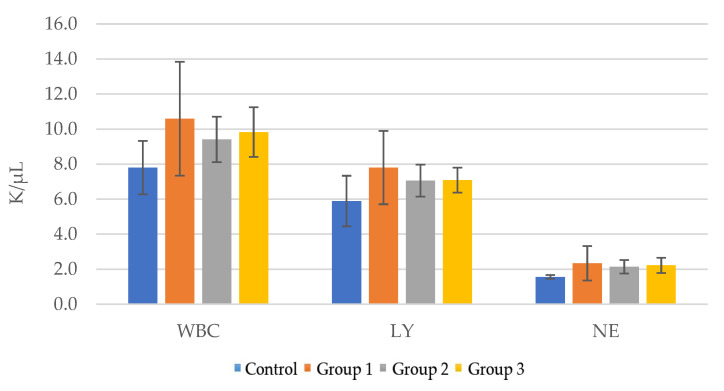
Evaluation of the number of leukocytes in the groups that received the novel nutraceutical compared to the control group: total numbers of leukocytes (WBC), lymphocytes (LY) and neutrophils (NE) in mice.

**Figure 12 pharmaceutics-15-01292-f012:**
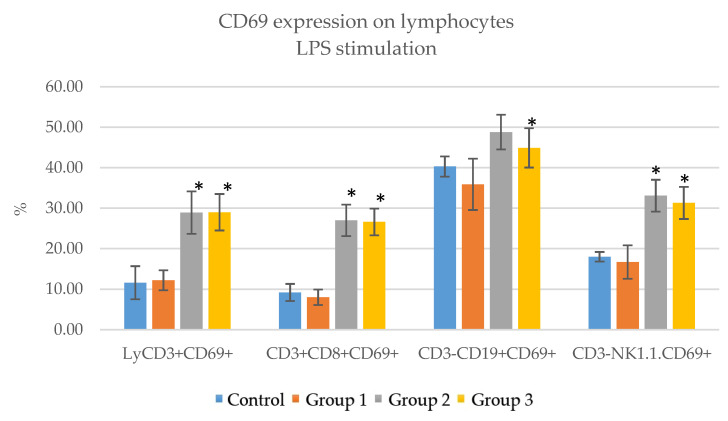
CD69 expression on lymphocytes in the groups that received ImunoBoost compared to the control group—stimulation with LPS; * = *p* value < 0.05.

**Figure 13 pharmaceutics-15-01292-f013:**
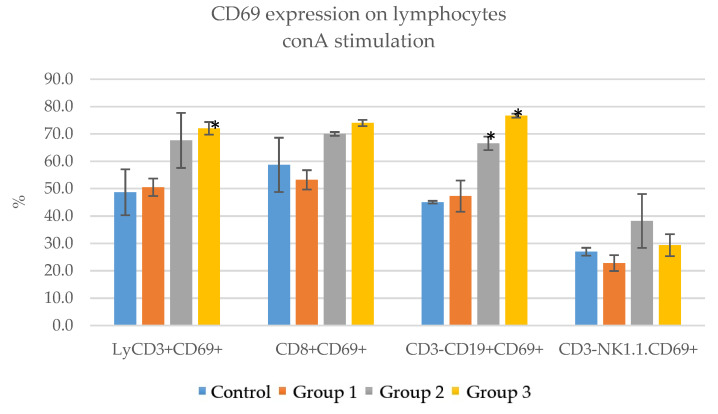
CD69 expression on lymphocytes in the groups that received ImunoBoost compared to the control group—stimulation with conA; * = *p* value < 0.05.

**Figure 14 pharmaceutics-15-01292-f014:**
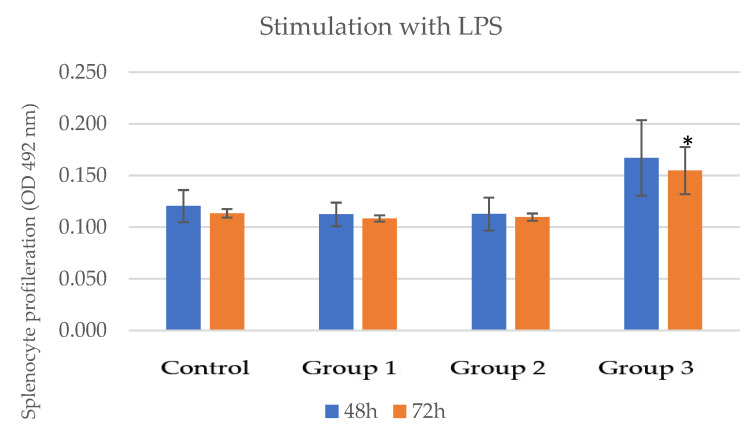
Lymphocyte proliferation capacity in groups that received the novel nutraceutical compared to the control group—stimulation with LPS; * = *p* value < 0.05.

**Figure 15 pharmaceutics-15-01292-f015:**
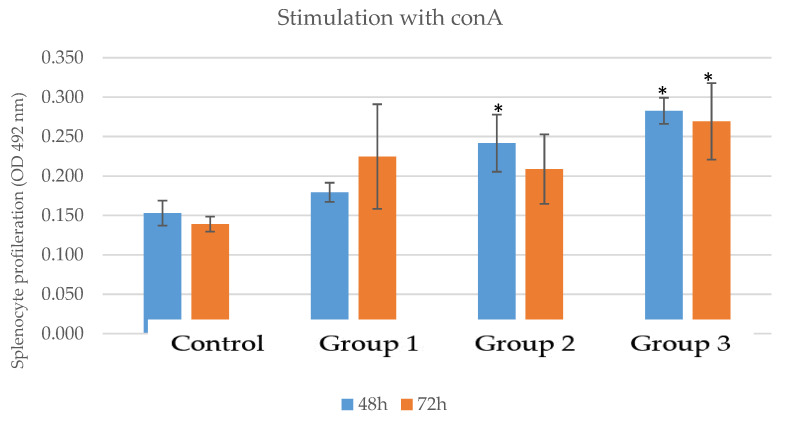
Lymphocyte proliferation capacity levels in the groups that received the novel nutraceutical compared to the control group—stimulation with conA; * = *p* value < 0.05.

**Table 1 pharmaceutics-15-01292-t001:** Monoclonal antibodies used for lymphocyte immunophenotyping.

Monoclonal Antibody	Fluorocrom	Isotype	Clone
anti-mouse CD3ɛ	Alexa Fluor 647	Armenian Hamster IgG	145-2C11
anti-mouse CD4	PE/Cy7	Rat IgG2b, κ	GK1.5
anti-mouse CD8a	Alexa Fluor 488	Rat IgG2a, κ	53-6.7
anti-mouse CD19	PerCP/Cy5.5	Rat IgG2a, κ	6D5
anti-mouse NK1.1	PE	Mouse IgG2a, κ	PK136

**Table 2 pharmaceutics-15-01292-t002:** Monoclonal antibodies used for the CD69 marker expression analysis.

Monoclonal Antibodies	Fluorocrom	Isotype	Clone
anti-mouse CD3ɛ	Alexa Fluor 647	Armenian Hamster IgG	145-2C11
anti-mouse CD69	PE/Cy7	Armenian Hamster IgG	H1.2F3
anti-mouse CD8a	Alexa Fluor 488	Rat IgG2a, κ	53-6.7
anti-mouse CD19	PerCP/Cy5.5	Rat IgG2a, κ	6D5
anti-mouse NK1.1	PE	Mouse IgG2a, κ	PK136

**Table 3 pharmaceutics-15-01292-t003:** Bioburden assessment of the novel nutraceutical.

Sample	Evaluation	Result
ImunoBoost	TAMC *, CFU/g	<10 CFU/g
	TYMC *, CFU/g	<10 CFU/g

* Total number of aerobic bacteria (TAMC) and total number of yeasts and molds (TYMC).

**Table 4 pharmaceutics-15-01292-t004:** Results obtained for selected elements reported as mean values ± SD, n = 3.

As (mg/kg)	Cd (mg/kg)	Hg (mg/kg)	Pb (mg/kg)
0.0312 ± 0.12	0.0171 ± 0.03	n.d	0.0201 ± 0.08

**Table 5 pharmaceutics-15-01292-t005:** Effect of administering a 2000 mg/Kg body dose of novel nutraceutical on body weight (g) in mice—acute toxicity testing.

	Body Weight/g		
Group	0 Days	7 Days	14 Days
Control	22.70 ± 0.3	22.73 ± 0.4	23.27 ± 0.4
Acute dose	22.10 ± 0.4	22.10 ± 0.2	22.20 ± 1.1

**Table 6 pharmaceutics-15-01292-t006:** Effect of administrating a dose of 2000 mg/Kg body weight of nutraceutical on the organ weight index (mg/g) in mice—acute toxicity testing.

Group	Thymus Index	Spleen Index	Liver Index	Left Kidney Index	Right Kidney Index
Control	1.0 ± 0.6	3.0 ± 0.4	44.0 ± 2.5	4.7 ± 0.5	5.4 ± 1.0
Acute dose	1.4 ± 0.1	2.7 ± 0.5	48.2 ± 3.5	5.3 ± 0.0	5.7 ± 0.4

**Table 7 pharmaceutics-15-01292-t007:** Effect of administrating a dose of 2000 mg/Kg of nutraceutical on the total numbers of leukocytes (WBC), lymphocytes (LY), neutrophils (NE) and monocytes (MO) in mice—acute toxicity testing; * *p = p* value.

Group	WBC (K/μL)	LY (K/μL)	NE (K/μL)	MO (K/μL)
Control	7.8 ± 1.52	5.89 ± 1.44	1.56 ± 0.11	0.33 ± 0.4
Acute dose	12.67 ± 0.49	9.76 ± 0.48	2.36 ± 0.29	0.42 ± 0.02
* *p*	0.0062	0.0107	0.0116	0.0703

**Table 8 pharmaceutics-15-01292-t008:** Effect of the administration of a novel nutraceutical in three different concentrations on the body weight (g) of mice.

	Body Weight/g			
Group	0 Days	7 Days	14 Days	21 Days
Control	22.30 ± 0.7	22.70 ± 0.3	22.73 ± 0.4	23.27 ± 0.4
Group 1 (50 mg/Kg)	21.23 ± 0.1	21.67 ± 0.2	21.73 ± 0.3	21.73 ± 0.4
Group 2 (100 mg/Kg)	21.27 ± 0.2	21.53 ± 0.2	21.30 ± 0.8	22.17 ± 0.6
Group 3 (200 mg/Kg)	22.50 ± 0.3	22.17 ± 0.4	22.40 ± 0.3	22.33 ± 0.4

**Table 9 pharmaceutics-15-01292-t009:** Effect of administration of novel nutraceutical in three different concentrations on the organ weight index (mg/g) in mice.

Group	Thymus Index	Spleen Index	Liver Index	Left Kidney Index	Right Kidney Index
Control	1.0 ± 0.6	3.0 ± 0.4	44.0 ± 2.5	4.7 ± 0.5	5.4 ± 1.0
Group 1 (50 mg/Kg)	1.4 ± 0.5	3.7 ± 0.4	51.7 ± 4.6	5.7 ± 0.6	5.4 ± 1.1
Group 2 (100 mg/Kg)	1.2 ± 0.3	2.7 ± 0.4	45.9 ± 3.0	5.4 ± 0.6	5.1 ± 0.8
Group 3 (200 mg/Kg)	1.2 ± 0.3	3.1 ± 0.5	45.4 ± 0.9	4.9 ± 0.4	4.8 ± 0.3

**Table 10 pharmaceutics-15-01292-t010:** Effect of the administration of a novel nutraceutical in three different concentrations on the total numbers of leukocytes (WBC), lymphocytes (LY), neutrophils (NE) and monocytes (MO) in mice.

Group	WBC (K/μL)	LY (K/μL)	NE (K/μL)	MO (K/μL)
Control	7.8 ± 1.5	5.89 ± 1.4	1.56 ± 0.1	0.33 ± 0.1
Group 1 (50 mg/Kg)	10.59 ± 3.2	7.80 ± 2.1	2.34 ± 1	0.31 ± 0.1
Group 2 (100 mg/Kg)	9.41 ± 1.3	7.05 ± 0.9	2.14 ± 0.4	0.17 ± 0.1
Group 3 (200 mg/Kg)	9.83 ± 1.4	7.08 ± 0.7	2.22 ± 0.4	0.26 ± 0.0

**Table 11 pharmaceutics-15-01292-t011:** The effect of administering ImunoBoost in three different concentrations on the lymphocyte populations and their subpopulations in mice.

Group	LyCD3+ Mean ± SD	CD3+CD8+ Mean ± SD	CD3+CD4+ Mean ± SD	CD3−CD19+ Mean ± SD	CD3−NK1.1.+ Mean ± SD	Th/Ts Mean ± SD
Control	50.4 ± 4.6	41.6 ± 1.5	52.8 ± 1.5	32.6 ± 3.7	3.2 ± 0.3	1.3 ± 0.1
Group 1 (50 mg/Kg)	51.3 ± 4.6	41.7 ± 2.6	52.5 ± 2.8	35.5 ± 4.2	3.2 ± 0.6	1.3 ± 0.2
Group 2 (100 mg/Kg)	51.3 ± 6.7	40.8 ± 1.4	53.4 ± 1.8	31.3 ± 7.2	3.1 ± 0.3	1.3 ± 0.2
Group 3 (200 mg/Kg)	50.4 ± 1.9	41.9 ± 0.5	52.1 ± 0.5	31.9 ± 5.7	3.3 ± 0.5	1.2 ± 0.02

**Table 12 pharmaceutics-15-01292-t012:** The effect of administering ImunoBoost in three different concentrations on the expression of the CD69 marker on lymphocyte surfaces in mice—stimulation with LPS; * *p = p* value.

Group	LyCD3+CD69+		CD8+CD69+		CD3−CD19+CD69+		CD3−NK1.1.CD69+	
	Mean ± SD	* *p*	Mean ± SD	* *p*	Mean ± SD	* *p*	Mean ± SD	* *p*
Control	11.6 ± 4.1		9.2 ± 2.1		40.3 ± 2.5		18 ± 1.2	
Group 1 (50 mg/Kg)	12.2 ± 2.5	0.8557	8 ± 1.9	0.5048	35.9 ± 6.3	0.3231	16.7 ± 4.1	0.6296
Group 2 (100 mg/Kg)	28.9 ± 5.2	0.0108	27 ± 3.9	0.0021	48.8 ± 4.3	0.0416	33.1 ± 3.9	0.0031
Group 3 (200 mg/Kg)	29 ± 4.5	0.0077	26.6 ± 3.3	0.0015	44.9 ± 4.9	0.224	31.3 ± 4	0.0051

**Table 13 pharmaceutics-15-01292-t013:** The effect of administering ImunoBoost in three different concentrations on the expression of the CD69 marker on lymphocyte surfaces in mice—stimulation with conA; * *p = p* value.

Group	LyCD3+CD69+		CD8+CD69+		CD3−CD19+CD69+		CD3-NK1.1.CD69+	
	Mean ± SD	* *p*	Mean ± SD	* *p*	Mean ± SD	* *p*	Mean ± SD	* *p*
Control	48.7 ± 8.4		58.7 ± 9.9		45.1 ± 0.5		27 ± 1.4	
Group 1 (50 mg/Kg)	50.5 ± 3.2	0.7398	53.2 ± 3.5	0.4197	47.3 ± 5.7	0.5395	22.8 ± 2.9	0.0895
Group 2 (100 mg/Kg)	67.6 ± 10.1	0.066	70 ± 0.7	0.1202	66.6 ± 2.5	0.0001	38.2 ± 9.8	0.1223
Group 3 (200 mg/Kg)	72.1 ± 2.3	0.0094	74 ± 1.1	0.0567	76.7 ± 0.7	<0.0001	29.36 ± 4	0.383

**Table 14 pharmaceutics-15-01292-t014:** Effect of the administration of the novel nutraceutical in three different concentrations on mouse lymphocyte proliferation—stimulation with LPS.

Group	Culture of 48 h			Culture of 72 h		
	OD	* *p*	Proliferation Index	OD	* *p*	Proliferation Index
Control	0.12 ± 0.016			0.113 ± 0.004		
Group 1 (50 mg/Kg)	0.112 ± 0.011	0.5109	0.93	0.108 ± 0.003	0.1688	0.96
Group 2 (100 mg/Kg)	0.113 ± 0.016	0.5826	0.94	0.110 ± 004	0.3083	0.97
Group 3 (200 mg/Kg)	0.167 ± 0.037	0.1112	1.39	0.155 ± 0.023	0.037	1.36

* OD = optical density.

**Table 15 pharmaceutics-15-01292-t015:** Effect of the administration of the novel nutraceutical in three different concentrations on mice lymphocyte proliferation—conA stimulation.

Group	Culture of 48 h			Culture of 72 h		
	OD	* *p*	Proliferation Index	OD	* *p*	Proliferation Index
Control	0.153 ± 0.016			0.139 ± 0.01		
Group 1 (50 mg/Kg)	0.179 ± 0.012	0.0842	1.17	0.225 ± 0.066	0.0913	1.62
Group 2 (100 mg/Kg)	0.242 ± 0.036	0.0178	1.58	0.209 ± 0.044	0.0552	1.5
Group 3 (200 mg/Kg)	0.283 ± 0.017	0.0006	1.85	0.269 ± 0.049	0.0102	1.94

* OD = optical density.

## Data Availability

The data presented in this study is available at links associated with each reference and within manuscript.

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
