# Peer review of "In Vivo Acute Toxicity and Immunomodulation Assessment of a Novel Nutraceutical in Mice"

_pharmaceutics, 2023, doi:10.3390/pharmaceutics15041292_

Round 1

Reviewer 1 Report

The concept of immunoceuticals refers to any nutraceuticals that are able to provide beneficial immunomodulatory actions that support and bolster the optimal immune system functioning. Since our immune functions are indispensable in defending the body against pathogens, diseases, and other external attacks, therefore it requires the functional foods and novel nutraceuticals. Immunoceuticals are considered effective in improving immune functions and reducing the incidence of immunological disorders, and therefore the main focus of this study was to assess the immunomodulatory properties and possible acute toxicity of a novel nutraceutical with actives of natural origin on mice for 21 days. The potential hazards of the novel nutraceutical were investigated and addressed the acute toxicity on mice for 21 days by determining body and organ indexes, leukocyte analysis, flow cytometry immunophe-notyping of lymphocytes, and expression of the CD69 activation marker. Results obtained indicate an increased number of lymphocytes, as well as activation, and proliferation post-nutraceutical administration. The manuscript is interesting, well designed, the experiments were performed carefully and I can suggest the acceptance of the manuscript.

Author Response

Thank you for reviewing our manuscript. We gratefully appreciate your feedback. 

Reviewer 2 Report

I suggest the title should be rewritten to a rather elegant form. Time of exposure is not necessary to be mentioned, and "solid oral dosage" either. Since the brand new product is already patented by the group, an striking title could attract readers to the main findings of this study, the same as for the Abstract.

Most of abstract is devoted to the background of the work, whereas only the last 6 lines brings some briefing design and results without any conclusion. Thus, I recommend the abstract must be rewritten to highlight the major findings, impact and further implications.

In Material and Methods, subsection 2.1, is the patent number or deposit protocol available? It is important to have one of them stated in the manuscript to avoid lack of novelty during patent submission process; a publication prior to the end of this step can compromise the acceptance of document for deposit.

In Material and Methods, subsection 2.2, I suggest to separate a topic for animals and ethic concerns, acute toxicity and subacute/immunomodulatory assessment. Regulatory guidelines concerning those tests must be cited in order to adequately document the investigation, such as OECD, EMA and/or FDA guidelines. Moreover, are samples obtained from acute toxicity tests submitted to protocols of following subtopics? Therefore it is important to separate those protocols (acute and subacute/immunomodulatory).

Additionally, there is some ethics concern regarding blood collection and euthanasia (not sacrifice! please change the term). Was the collection of blood from retro-orbital plexus performed under anaesthesia? If yes, please describe. Furthermore, were animals euthanized under anaesthesia? Please describe this section based on the best practices in animal handling. ARRIVE guidelines brings a suitable guideline (https://arriveguidelines.org/).  

The Discussion is very poor and do not discuss the relevance of findings regarding to a real immunomodulation occuring, focusing on a potential applications. I understand those findings are important. However, the manuscript is not exploring well in the present form. Please rethink discussion and explore deeply the findings. Most of important results are comprised between lines 580 and 599, and only three references are cited.

Author Response

Dear Reviewer

We thank you for your time, effort, and suggestions. Here are our responses to your observations:

  1. I suggest the title should be rewritten to a rather elegant form. Time of exposure is not necessary to be mentioned, and "solid oral dosage" either. Since the brand new product is already patented by the group, an striking title could attract readers to the main findings of this study, the same as for the Abstract.

We revised the title and the abstract.

  1. Most of abstract is devoted to the background of the work, whereas only the last 6 lines brings some briefing design and results without any conclusion. Thus, I recommend the abstract must be rewritten to highlight the major findings, impact and further implications.

The abstract was rewritten as recommended.

  1. In Material and Methods, subsection 2.1, is the patent number or deposit protocol available? It is important to have one of them stated in the manuscript to avoid lack of novelty during patent submission process; a publication prior to the end of this step can compromise the acceptance of document for deposit.

Thank you for pointing out this important aspect. We clarified it and mentioned the depository number of the patent.

  1. In Material and Methods, subsection 2.2, I suggest to separate a topic for animals and ethic concerns, acute toxicity and subacute/immunomodulatory assessment. Regulatory guidelines concerning those tests must be cited in order to adequately document the investigation, such as OECD, EMA and/or FDA guidelines. Moreover, are samples obtained from acute toxicity tests submitted to protocols of following subtopics? Therefore it is important to separate those protocols (acute and subacute/immunomodulatory).

We separated acute toxicity protocols from the immunomodulatory protocols and specified the exact guidelines followed as well as the ethics committee approval for testing.

  1. Additionally, there is some ethics concern regarding blood collection and euthanasia (not sacrifice! please change the term). Was the collection of blood from retro-orbital plexus performed under anesthesia? If yes, please describe. Furthermore, were animals euthanized under anesthesia? Please describe this section based on the best practices in animal handling. ARRIVE guidelines brings a suitable guideline (https://arriveguidelines.org/).  

Prior to blood collection, all animals were anesthetized with a ketamine/acepromazine cocktail, ketamine 100 mg/kg (ketamine 10%, Medistar Arzneimittelvertrieb Gmbh, Ascheberg, Germany), and acepromazine 5 mg/kg (Calmivet Solution Injectable Acepromazine 5 mg, Vétoquinol SA, Lure, France). After blood samples were collected, the animals were euthanized by dislocation of the cervical spine, and organs of interest were harvested.

  1. The Discussion is very poor and do not discuss the relevance of findings regarding to a real immunomodulation occurring, focusing on a potential applications. I understand those findings are important. However, the manuscript is not exploring well in the present form. Please rethink discussion and explore deeply the findings. Most of important results are comprised between lines 580 and 599, and only three references are cited.

Thank you. We revised the discussion section and focused on the relevance of our findings.

Reviewer 3 Report

The article describes immunimodulatory effect and acute toxicity of a new nutraceutical Imunoboost. 

It is mostly nicely written (I suggest grammar and spelling check, for example "weekly" instead of weakly"). Decimal numbers should be written with dots, not commas.

Presentation of the Tables should be improved. Each Table should be clear by itself, and it is not always the case. 

For example, what is TAMC in Table 3? That should be explained in the Table legend. 

Table 5 and 8 - please include "body weight/g" in the table, above "0 days", "7 days", "14 days".

Explain what is spleen index, liver index etc (Table 6 and 9). The title states"the effect of ..... on the body weight index..."

Table 7 and 10 - what is "K"??

Table 11, 12, 13 - what do numbers represent? Please include the value you measured in the Table.

Table 14 and 15  - what is OD and proliferation index?  Please explain what you measured.

In some tables authors write the dose of nutraceutical, and in some it is missing.

Discussion - last paragraph, last sentence - statement is too bold and exaggerated. This is a  scientific article, and the authors should stick to their finding. There is no evidence that nutraceutical "strengthens organism's predisposition to the seasonal flue, boosts immunity by its antioxidant potential, alleviates tiredness and fatigue, and propels the normal energy-yielding metabolism." It sounds as a commercial, not a conclusion.

Author Response

Dear Reviewer

We thank you for your time, effort, and suggestions. Here are our responses to your observations:

The article describes immunomodulatory effect and acute toxicity of a new nutraceutical ImunoBoost

It is mostly nicely written (I suggest grammar and spelling check, for example "weekly" instead of weakly"). Decimal numbers should be written with dots, not commas.

We performed a grammar and spelling check.

Presentation of the Tables should be improved. Each Table should be clear by itself, and it is not always the case. For example, what is TAMC in Table 3? That should be explained in the Table legend. 

We improved the presentation of the tables and harmonized them. In table 3 TAMC represents the total number of viable aerobic microorganisms. We mentioned this abbreviation in section 2.1.1 Bioburden assessment, but we also included the explanation of the abbreviation at lines 387 – 388 as well in the table legend.

Table 5 and 8 - please include "body weight/g" in the table, above "0 days", "7 days", "14 days".

We included it.

Explain what is spleen index, liver index etc (Table 6 and 9). The title states "the effect of ..... on the body weight index..."

We corrected the table explanation. We performed anthropometric measurements of organs after the administration of nutraceutical. The index was determined by comparison of the control group with the treated group measurements according to formula (1) Organ index (mg/g) = organ weight (mg)/body weight (g) from section 2.3.2.

Table 7 and 10 - what is "K"??

K/µL means Thousand/µL

Table 11, 12, 13 - what do numbers represent? Please include the value you measured in the Table.

For CD3+ lymphocytes (LyCD3+), the results are presented as percentages from lymphocytes. For T, B, and NK cells, the results are presented as percentage of CD3ε+ or CD3ε- lymphocytes. All data are presented as mean ± SD and we included it in the table.

Table 14 and 15  - what is OD and proliferation index?  Please explain what you measured.

OD = Optical density. We measured mouse lymphocyte proliferation. The proliferation index was calculated with the formula from section 2.3.6 Proliferation index = OD of the treated group / OD of the control group using the CellTiter 96® AQueous One Solution Cell Proliferation Assay (Promega).

In some tables authors write the dose of nutraceutical, and in some it is missing.

We harmonized this aspect in all tables.

Discussion - last paragraph, last sentence - statement is too bold and exaggerated. This is a scientific article, and the authors should stick to their finding. There is no evidence that nutraceutical "strengthens organism's predisposition to the seasonal flue, boosts immunity by its antioxidant potential, alleviates tiredness and fatigue, and propels the normal energy-yielding metabolism." It sounds as a commercial, not a conclusion.

We improved the last statement in accordance with our findings.    

Round 2

Reviewer 2 Report

Corrections were performed as requested. The manuscript is suitable to be accepted for publication.